# Convergent genomic signatures of domestication in sheep and goats

Florian J. Alberto et al.[#]

The evolutionary basis of domestication has been a longstanding question and its genetic architecture is becoming more tractable as more domestic species become genome-enabled. Before becoming established worldwide, sheep and goats were domesticated in the fertile crescent 10,500 years before present (YBP) where their wild relatives remain. Here we sequence the genomes of wild Asiatic mouflon and Bezoar ibex in the sheep and goat domestication center and compare their genomes with that of domestics from local, traditional, and improved breeds. Among the genomic regions carrying selective sweeps differentiating domestic breeds from wild populations, which are associated among others to genes involved in nervous system, immunity and productivity traits, 20 are common to *Capra* and *Ovis*. The patterns of selection vary between species, suggesting that while common targets of selection related to domestication and improvement exist, different solutions have arisen to achieve similar phenotypic end-points within these closely related livestock species.

[#]A full list of authors and their affliations appears at the end of the paper. Florian J. Alberto, Frédéric Boyer, Pablo Orozco-terWengel and Ian Streeter contributed equally to this work.

Plant and animal domestication represented a major turning point in human evolution, leading to the emergence of farming during the Neolithic[1]. By providing a series of independent long-term evolutionary experiments where plants and animals were selected for specific traits, this process has been of longstanding interest to evolutionary biologists including Darwin[2]. Domestic species share many morphological, behavioral, and physiological traits[3], collectively referred to as domestication syndromes. In animals, selection for tameness[4], changes in development rates[5–7], and developmental pathways[8–10] are hypothesized to have triggered domestication and the unintentional emergence of domestication syndrome-related characters such as piebald coat color and lop ears. Following the attainment of tame animals, deliberate selection for improved phenotypes related to primary (e.g., meat or milk) and secondary (e.g., stamina or speed) domestication products took place. While domestication triggered positive selection for many traits in domestic species, it also led to the relaxation of selection for traits of reduced importance in domestic conditions (such as camouflage coloration, twinning, sexual selection, and predator avoidance[4,11]). Recently, genome-wide analyses have identified a number of variants that differentiate domesticates from their wild counterparts including in species such as chicken[12], pig[13], dog[14], rabbit[15], cattle[16], and horse[17]. However, evidence for trans-specific signatures of domestication remains largely unexplored and support for common genes related to domestication or subsequent improvement across domestic animals remains elusive. This might reflect that selection acted on species-specific traits during domestication or that domestication traits are predominantly polygenic and/or pleiotropic in nature, allowing selection to target different genes while resulting in similar phenotypes (e.g., polledness is driven by different genes in sheep and goat[18,19]).

To test for common, trans-specific signatures of selection, we took advantage of the parallel history of domestication in the closely related sheep (*Ovis aries*) and goat (*Capra hircus*). Their wild ancestors, the Asiatic mouflon (*Ovis orientalis*) and the Bezoar ibex (*Capra aegagrus*) diverged during the late Miocene[20], and were domesticated ~10.5 kya (thousand years ago) in the same region of the Middle-East (South-eastern Anatolia and the Iranian Zagros Mountains)[21,22]. Since then, humans have spread domestic sheep and goats beyond their native range, and ultimately throughout the world. Importantly, unlike other common livestock there is no evidence that they hybridized with native wild relatives in the diffusion process out of the domestication center[23], which facilitates the investigation of the genomics changes underlying domestication in different environments and production systems.

Herein we sequence the genomes of wild Asiatic mouflon and Bezoar ibex in the sheep and goat domestication center and compare their genomes with that of domestics from local, traditional, and improved breeds, using haplotype differentiation as the signature of selection. Wild and domestic groups support selection for a total of 90 regions, out of which functional annotations are available for 59, based on overlapping or close genes. Interestingly, 20 regions are common to *Capra* and *Ovis* and exhibit patterns of selection that vary between species. This suggests that while common targets of selection related to domestication and improvement exist, different solutions have arisen to achieve similar phenotypic end-points.

## Results

**Sampling design**. To identify genomic regions associated with sheep and goat domestication, for both, we generated and analyzed genome data from wild representatives and three domestic groups in both species (Fig. 1a). In total, we generated high-quality (12–14 fold coverage) genome sequences from 13 wild Asiatic mouflon (IROO) and 18 Bezoar ibex (IRCA), and 40 sheep and 44 goats, representing two groups of traditionally managed populations. The first domestic group was from Iran (IROA: 20 sheep, IRCH: 20 goats), designed to survey animals found within the geographic envelope of the domestication center, sympatric with their wild counterparts. The second domestic group was from Morocco (MOOA: 20 sheep, MOCH: 20 goats), located at the terminal end of the Southern Mediterranean colonization route[24]. The third domestic group comprised a worldwide panel of mostly industrial breeds (wpOA: 20 sheep, wpCH: 14 goats), which we expected to have experienced stronger selection and more complex demographic histories. Thus, our nested sampling was designed to distinguish candidates shared by all domestic groups from signatures of local adaptation in traditionally managed populations (Iran and Morocco) or in more recently intensively selected breeds (worldwide panel), in a replicated manner for both sheep and goat.

**Global patterns of genomic diversity**. We identified about 33 million and 23 million single nucleotide polymorphisms (SNPs) in *Ovis* and *Capra*, respectively (Supplementary Note 1). Interestingly, Bezoar ibex showed lower nucleotide diversity than Iranian goats and higher inbreeding than Iranian and Moroccan goats (Supplementary Table 1). In contrast, nucleotide diversity was higher in Asiatic mouflon than in domestic sheep. We inferred higher genetic load in *Ovis* than *Capra* (Supplementary Table 1). Genetic load was higher in sheep than in mouflon with a significant increase in the domestic world panel, while in *Capra* the load was instead significantly higher for wild individuals (Supplementary Table 1 and Supplementary Fig. 1). The inbreeding coefficient $F$ was positively correlated with the genetic load per homozygous position (Supplementary Fig. 1; Pearson correlation coefficient $r > 0.87$ and $p$-value $< 10^{-23}$ for both genera). Analysis of relaxation of functional constraints related to domestication was conducted only for *Ovis*, as the high genetic load observed for the IRCA group precluded this investigation. We found 277 genes with significantly higher deleterious load in domestic sheep than in Asiatic mouflon (Supplementary Data 1). Enrichment analysis revealed that these genes are mostly involved in morphological changes, including adipogenesis, anatomical structure, severe short stature, and cervical subluxation (Supplementary Table 2, adjusted $p$-value $\leq 0.01$).

When tracing the demographic history using multiple sequentially Markovian coalescent (MSMC)[25], wild and domestic groups of the same genus showed similar effective sizes prior to domestication (~10.5 kya) as expected with their common origin. *Capra* and *Ovis* demonstrated different effective sizes but showed similar patterns of fluctuation. At the time of domestication, the size of wild populations remained stable or increased while the effective size for domestic groups subsequently decreased, after an initial period of growth for goats only. During the last two millennia, wild populations declined while domestic groups increased (see Supplementary Note 2 and Supplementary Fig. 2).

Genetic structure analysis performed with sNMF[26] within *Ovis* and *Capra* groups showed two isolated gene pools representing wild and domestic animals for both sheep and goat (Fig. 1b, c). Using Treemix[27] and f3 statistics[28], we could not detect evidence for recent hybridization between wild and domestic animals in either genus (see Supplementary Note 3, Supplementary Fig. 3, Supplementary Table 3, and Supplementary Data 2), facilitating further comparisons aimed at detecting signatures of selection.

**Patterns of selection.** Using haplotype differentiation as the signature of selection[29] and then applying a stratified FDR framework (see Methods and Supplementary Fig. 4), we found 46 and 44 candidate regions under selection in *Ovis* and *Capra*, respectively (Fig. 2a, Supplementary Note 4 and Supplementary Table 4). The pattern of haplotype clustering was similar among the three domestic groups in all such regions (Supplementary Fig. 5). Comparisons of nucleotide diversity and haplotype clustering between wild and domestic groups supported directional positive or stabilizing selection for a total of 45 regions in sheep and 30 regions in goats, with the remaining 15 being inferred as having undergone relaxed or diversifying selection. Out of these 90 regions, functional annotations are available for 59, based on overlapping or close genes (Supplementary Note 4 and Supplementary Data 3), which displayed pleiotropic effects. Interestingly the representation of the higher level GO terms for these genes under selection differed from those of the reference build from the Uniprot database ($\chi^2$-test, *p*-value $\leq 0.05$) due to an over-representation of genes related to pigmentation and, to a lesser extent, in biological adhesion and rhythmic processes (Supplementary Data 4). In livestock, most of these genes have already been associated to phenotypic effects related to immunity (14 genes), productivity traits associated to milk composition (11 genes), meat (11 genes), and hair characteristics (4 genes), fertility (2 genes), and neural development, and the nervous system (5 genes) (Supplementary Data 3). Most of the 1076 variants detected in both genera showed selection signatures in non-coding sequences (36% intergenic, 50% intronic, and 14% in up- and downstream positions plus three exonic changes—two missense and one nonsense). For *Capra*, we found a significant enrichment for intronic, upstream gene, and downstream gene regions (Supplementary Table 5).

Importantly, the stratified FDR approach showed convergence (i.e., shared signals of selection) between both genera, as in homologous regions the proportion of significant SNPs found under selection in *Ovis* increased with the stringency for detecting selection in *Capra* and vice versa (see Methods and Fig. 2b). Twenty candidate regions for selection were common to both genera (Fig. 2a and Supplementary Data 3). As for genus-specific regions these were associated with genes involved in the nervous system, immunity and several improvement traits (Table 1). Noticeably, among these genes, *KITLG* also presented a higher genetic load in sheep than in Asiatic mouflon (Supplementary Data 1), as a possible result of strong selection in domestics.

## Discussion

Genomic signatures related to domestication and/or improvement were found both in response to demographic and selective differences between wild and domestic populations. *Capra* and *Ovis* showed opposite global patterns of genomic diversity. In *Capra*, the low nucleotide diversity and high inbreeding in the Bezoar ibex compared to goats has already been documented[30]. This observation could result from the different demographic trajectories of wild and domestic populations comprising the recent severe decline of wild populations resulting from extensive poaching and habitat fragmentation[31]. These differences could also explain the higher genetic load in Bezoar ibex. In *Ovis*, wild populations are more diverse than their domestic counterparts, which could be due to the lower effective size in the domestics observed between 10 and 1.5 kya (Supplementary Fig. 2). The increased genetic load in sheep may represent a domestication signature, where demographic bottlenecks reduced the efficacy of

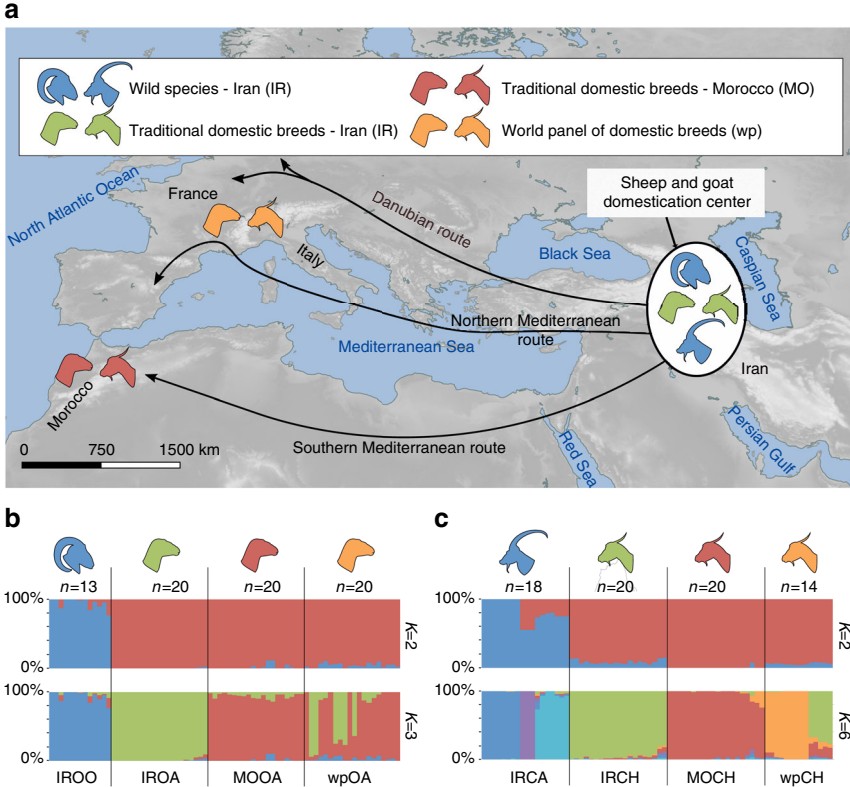

**Fig. 1** Sampling with regard to the domestication center and main colonizations routes. **a** Sampling locations represented by the animal's silhouettes. The domestication center and the main colonization routes (Northern and Southern Mediterranean routes and Danubian route) are presented. Proportion of genomes assigned to *K* genetic clusters for *Ovis* (*K* = 2 and *K* = 3) (**b**) and *Capra* (*K* = 2 and *K* = 6) (**c**) individuals. The number of sampled individuals (*n*) is given

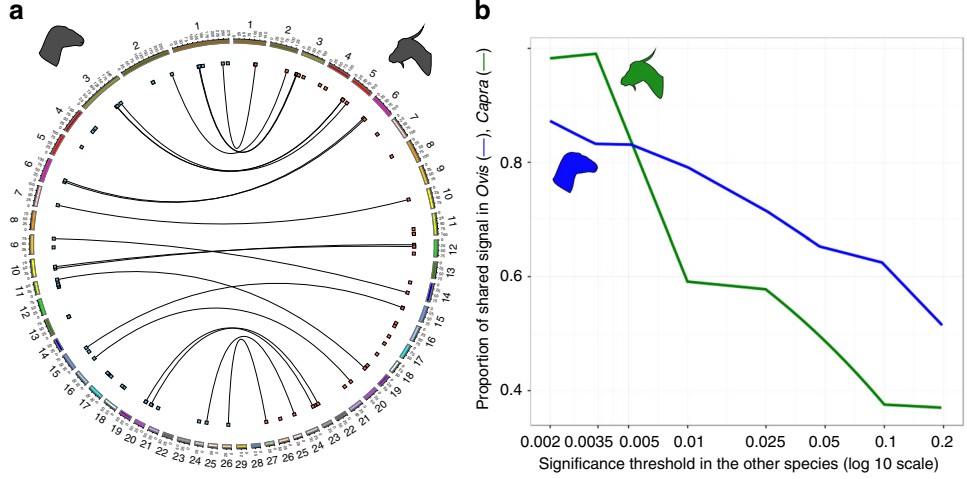

**Fig. 2** Specific and shared chromosomal regions selected during domestication. **a** Circular representation of the sheep (left) and goat (right) genomes. Dots represent regions under selection, lines indicate shared selected regions. Chromosomes sizes are in Megabases. **b** Convergence in the signals of selection between *Capra* and *Ovis*. In homologous genomic regions, the proportion of significant SNPs found under selection in *Ovis*, respectively *Capra* (y-axis), increases with the stringency for detecting selection in *Capra*, respectively *Ovis* (x-axis)

**Table 1 Homologous genomic regions differentiating wilds from domestics in *Ovis* and *Capra***

| Chromosome | Code | Gene | Δπ |
|---|---|---|---|
| Ovis 1 \| Capra 3 | *ENSOARG00000006800 \| SLAMF7* | Novel gene \| *SLAM* family member 7 | 0.12 \| 0.11 |
| Ovis 1 \| Capra 3 | *SLAMF1* | Signaling lymphocytic activation molecule precursor 1 | 0.13 \| 0.10 |
| Ovis 1 \| Capra 1 | Intergenic | None | 0.09 \| 0.14 |
| Ovis 2 \| Capra 2 | Intergenic | None | 0.15 \| 0.08 |
| Ovis 3 \| Capra 5 | *KITLG* | Proto-oncogene receptor tyrosine kinase ligand | 0.18 \| −0.15 |
| Ovis 3 \| Capra 5 | *KITLG* | Proto-oncogene receptor tyrosine kinase ligand | 0.26 \| −0.21 |
| Ovis 3\| Capra 5 | *HMGI-C* | High mobility group protein I-C | 0.11 \| 0.12 |
| Ovis 6 \| Capra 6 | *HERC5 \| HERC6* | *HECT & RLD* domain containing E3 ubiquitin protein ligase 5 \| 6 | 0.24 \| −0.11 |
| Ovis 6 \| Capra 6 | *SLC34A2* \| Intergenic | Solute carrier family 34 member 2 \| None | 0.18 \| 0.23 |
| Ovis 7 \| Capra 10 | Intergenic | None | 0.08 \| 0.20 |
| Ovis 9 \| Capra 14 | *POP1* | Ribonuclease P/MRP subunit | −0.03 \| 0.11 |
| Ovis 10 \| Capra 12 | *NBEA* | Neurobeachin | 0.11 \| 0.16 |
| Ovis 10 \| Capra 12 | *CRYL1* | Crystallin lambda 1 | 0.01 \| 0.19 |
| Ovis 11 \| Capra 19 | *RNF213* | Ring finger protein 213 | 0.13 \| −0.26 |
| Ovis 15 \| Capra 15 | *U1 \| HBE1* | U1 spliceosomal RNA \| Hemoglobin subunit epsilon-1 | 0.22 \| 0.09 |
| Ovis 16 \| Capra 20 | *TRIP13 \| SLC12A7* | Thyroid hormone receptor interactor 13 \| Solute carrier family 12 member 7 | 0.16 \| −0.16 |
| Ovis 20 \| Capra 23 | *SUPT3H* | *SPT3* homolog, *SAGA* and *STAGA* complex component | 0.08 \| 0.06 |
| Ovis 20 \| Capra 23 | *EXOC2 \| DUSP22* | Exocyst complex component 2 \| Dual specificity phosphatase 22 | 0.25 \| 0.03 |
| Ovis 24 \| Capra 25 | *HBM \| LUC7L* | Hemoglobin subunit Mu \| LUC like 7 | 0.33 \| 0.12 |
| Ovis 26 \| Capra 27 | *MTMR7* | Myotubularin related protein 7 | 0.15 \| 0.07 |

When different in both genera, information is given for *Ovis* | *Capra*. Positive Δπ indicates a lower diversity in domestics (e.g., directional positive or stabilizing selection in domestics) while negative values indicate a lower diversity in the wilds (e.g., diversifying selection / relaxation in domestics or recent positive selection in the wilds). The phenotypic effects presented are inferred from the bibliography and classified in a livestock perspective. The Uniprot GO terms associated to these genes are available from Supplementary Data 4

negative selection in purging deleterious mutations from the domestic gene pool. In both *Capra* and *Ovis* the tendency for a higher genetic load in world panels (which include industrial breeds) than in traditionally managed populations is concordant with such an impact of repeated bottlenecks, likely derived from intensive selection.

When describing patterns of selection, we found genomic signatures of selection shared between traditionally-managed domestic populations of sheep and goat, from both the domestication center (Iran), the terminal end of the Southern Mediterranean diffusion route (Morocco) and in more intensively selected breeds worldwide. The most parsimonious scenario involves selection in these genes before the divergence of these groups. However, this does not prejudge the time and localization of the selective events, which might have occurred during domestication or at an early improvement step in the fertile crescent, and/or also probably later on and elsewhere. Indeed, evidence exists that modern domestic populations are not directly related to the first domesticated animals due to population replacements e.g.,[32,33], or that nearly fixed domestic traits in modern populations are due to later Neolithic improvements[34]. The regions found under selection included both genes and genomic regions devoid of genes (i.e., 17 out of 90 regions). This could be due to a lack of functional annotation of these regions but also to selection targeting regulatory sequences. Regardless, almost all variants found to be under selection were in non-coding sequences, with only two missense mutations identified. Some of this non-genic signal might result from hitchhiking, e.g., to an unidentified causal mutation in coding regions, but could also reflect selection on regulatory sequences, since it has been

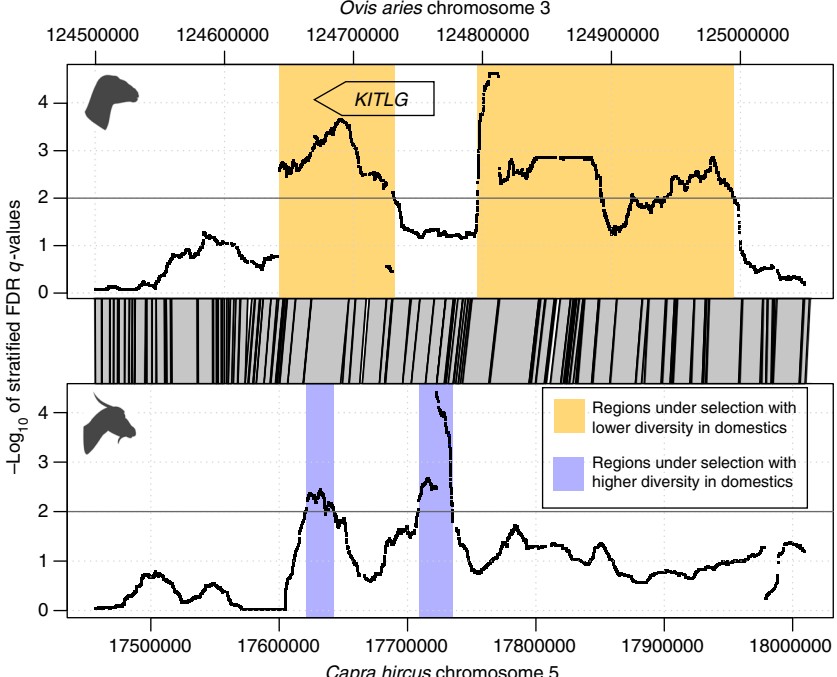

**Fig. 3** Chromosomal regions under selection overlapping the *KITLG* gene. Each genus bears two signatures of selection from which one comprises a part strictly homologous between genera. Both regions exhibit a lower diversity in domestics for *Ovis* and a higher diversity in domestics for *Capra*

shown that selective sweeps differentiating sheep from mouflon are enriched for coding genes and regulatory elements[35].

Importantly, 20 genomic regions were identified as being under selection in both *Ovis* and *Capra*. Interestingly, four genes show pleiotropic effects and have been related to phenotypic effects in several livestock species. *KITLG* has known associations with neural stem cell systems[36], coat color in mammals[37], and on litter size in goats[38]. *HMGI-C* is a major candidate for dwarfism in chickens[39] and *MTMR7* is involved in fatty acid composition in pigs[40]. *NBEA* is associated with wool crimping in sheep[41], but is also involved in neurotransmission[42] and may play a role on behavior in cattle[43]. Thus, the pleiotropic nature of these genes may have facilitated early domestication and/or subsequent improvement through behavioral changes and selection for productivity traits[8,10].

Of the 20 selection candidates common to sheep and goat, 14 selection signatures were congruent in both species. Interestingly, however, for *KITLG* (2 regions) and four other regions, we found evidence of different selective patterns between sheep and goat. Such contrasting signals may reflect complex spatio-temporal selection and multiple breeding strategies applied to these different traits. For example, for the pleiotropic *KITLG* gene, the divergent signals in sheep and goat (Fig. 3) could be explained by a relaxation of selection on coat color in goats, as already observed in horses[44] and pigs[9], whereas sheep have largely been selected for uniform fleece color (especially for white wool). Besides this, the loss of genome-wide diversity between wild and local domestic populations was only observed here for the mouflon-sheep comparison and not for goats, while inbreeding and genetic load tend to increase in sheep and decrease in goats. While contemporary demographic processes in wild populations may partially explain this observation, the complex nature of the selection signatures described here on pleiotropic genes challenges a simplistic view of the effects of domestication and improvement on the genomes of animal species. This contrasts with viewing domestication and subsequent artificial selection as a uniformly directed selective processes, focused on relatively canalized gene systems[45].

The combination of different patterns of selection involved in the domestication or subsequent improvement of livestock, along with the underlying pleiotropic gene systems detected suggest a more complex, multifaceted genetic response to a profound shift in the life histories of domestic animals. It is noteworthy that approximately half of the genes showing selection signatures in *Ovis*, show congruent signatures in *Capra*. This suggests that while common targets of selection exist within the genomes of these species, different solutions have arisen to achieve similar phenotypic selection goals.

## Methods

**Sampling**. Domestic sheep (*O. aries*) and goats (*C. hircus*) were sampled in Iran (IROA and IRCH groups, respectively) and Morocco (MOOA and MOCH groups, respectively) for a total of 20 animals per group (Supplementary Fig. 6). These samples were collected between January 2008 and March 2012 in the Northern part of Morocco and between August 2011 and July 2012 in North-Western Iran, in the frame of the Nextgen European project (Grant Agreement no. 244356) in accordance with ethical regulations of the European Union Directive 86/609/EEC. Ear-clips were collected from the distal part of the ear of randomly chosen animals, and immediately stored in 96% ethanol for one day before being transferred in silica-gel beads until DNA extraction.

The wild species Asiatic mouflon (*O. orientalis*) and Bezoar ibex (*C. aegagrus*) were sampled in North-western Iran within the domestication cradle[21,22]. Thirteen Asiatic mouflons and 18 Bezoar ibex tissues (respectively, IROO and IRCA groups, Supplementary Fig. 6) were collected either from captive or recently hunted animals, and from frozen samples available at the Iranian Department of Environment. This individual-based sampling approach is designed to minimize potential bias by avoiding the overrepresentation of local effects (e.g., local inbreeding).

**Additional data**. Additionally, a worldwide breed panel was assembled for sheep and goats (wpOA and wpCH, respectively). wpOA included 20 whole-genome re-sequencing (WGS) samples at 12x coverage representing 20 different worldwide breeds provided by the International Sheep Genome Consortium. wpCH consisted of 14 WGS samples sequenced at 12x coverage representing 9 European individuals, i.e., 2 French Alpine, and 2 French Saanen samples sequenced by INRA, 5 Italian Saanen samples provided by Parco Tecnologico Padano, and 5 Australian

individuals, i.e., 2 Boer, 2 Rangeland, and 1 Cashmere samples provided by the CSIRO (Supplementary Data 5).

**Production of WGS data.** Genomic DNA was successfully extracted from all tissue samples using the Macherey Nagel NucleoSpin 96 Tissue kit, adapting the manufacturer's protocol. Tissue sampling was performed in MN square-well blocks to obtain 25 mg fragments per sample. Three and a half MN square-96 blocks were prepared, and extraction was performed using a Tecan Freedom Evo Liquid handler following manufacturer's protocol. A pre-lysis step was carried out to homogenize samples with 180 µl of T1 Buffer and 25 µl of proteinase K overnight at 56 °C. To adjust binding conditions, 200 µl of BQ1 buffer were added and the sample plate was incubated 1 h at 70 °C; 200 µl of 100% ethanol were subsequently added. Lysates were transferred to Nucleospin Tissue binding plate and a vacuum (−0.2 bar, 5 min) was applied to remove the flow-through. Three washing steps were done with BW and B5 buffers, respectively, and a vacuum was applied again to discard the flow-through. Prior to the elution of genomic DNA, a Nucleospin Tissue binding plate silica membrane was dried under vacuum with at least −0.6 bar for 10 min. The elution step was performed with 100 µl of pre-warmed BE buffer (70 °C) and a centrifugation step at 3700 rcf for 5 min in 96-PCR wells. Genomic DNA was stored at 4 °C to avoid freeze-thawing and tested for concentration (as ng/µl) using the Picogreen method and using a Nanodrop.

Whole genomes were resequenced from 500 ng of genomic DNA that were sheared to a 150–700 bp range using the Covaris® E210 instrument for each sample and used for Illumina® library preparation by a semi-automatized protocol. End repair, A-tailing, and Illumina® compatible adaptors (BioScientific) ligation were performed using the SPRIWorks Library Preparation System and SPRI TE instrument (Beckmann Coulter) following the manufacturer's protocol. A 300–600 bp size selection was applied to recover most of the fragments. DNA fragments were amplified by 12 cycles of PCR using Platinum Pfx Taq Polymerase Kit (Life® Technologies) and Illumina® adapter-specific primers. Libraries were purified with 0.8x AMPure XP beads (Beckmann Coulter), and analyzed with the Agilent 2100 Bioanalyzer (Agilent® Technologies) and qPCR quantification. Libraries were sequenced with 100 base-length read chemistry in paired-end flow cell on the Illumina® HiSeq2000.

Illumina paired-end reads for *Ovis* were mapped to the sheep reference genome (OAR v3.1, GenBank assembly GCA_000298735.1[46]), and for *Capra* to the goat reference genome (CHIR v1.0, GenBank assembly GCA_000317765.1[47]), using BWA-MEM[48]. The BAM file produced for each individual was sorted using Picard SortSam and improved using sequentially Picard MarkDuplicates (http://picard.sourceforge.net), GATK RealignerTargetCreator and GATK IndelRealigner[49], and SAMtools calmd[50].

Variant discovery was carried out using three different algorithms: Samtools mpileup[50], GATK UnifiedGenotyper[51], and Freebayes[52]. Variant sites were identified independently for each of six groups, using the multi-sample modes of the calling algorithms: (i) 162 samples from MOOA; (ii) 20 samples from IROA; (iii) 14 samples from IROO; (iv) 162 samples from MOCH; (v) 20 samples from IRCH; (vi) 19 samples from IRCA. For some groups, the WGS of more individuals were available as part of the NextGen project (see above). The samples used in the present study were selected to obtain balanced groups of 20 individuals wherever possible. For IRCA and IROO groups, additional samples became available at a later stage and were added for downstream analyses. Animals with low alignment and calling quality were removed to obtain the final data set (Supplementary Data 5).

Within each group, there were two successive rounds of variant site quality filtering. Filtering stage 1 merged calls together from the three algorithms, whilst filtering out the lowest-confidence calls. A variant site passed if it was called by at least two different calling algorithms with phred variant quality >30. An alternate allele at a site passed if it was called by any one of the calling algorithms, and the genotype count was >0. Filtering stage 2 used Variant Quality Score Recalibration by GATK. First, we generated a training set of the highest-confidence variant sites within the group where (i) the site is called by all three variant callers with phred-scaled variant quality >100, (ii) the site is biallelic, (iii) the minor allele count is at least 3 while counting only samples with genotype phred-scaled quality >30. The training set was used to build a Gaussian model using the tool GATK VariantRecalibrator using the following variant annotations from UnifiedGenotyper: QD, HaplotypeScore, MQRankSum, ReadPosRankSum, FS, DP, InbreedingCoefficient. A Gaussian model was applied to the full data set, generating a VQSLOD (log odds ratio of being a true variant). Sites were filtered if VQSLOD <cutoff value. The cutoff value was set for each group by the following: Minimum VQSLOD = {the median value of VQSLOD for training set variants} −3 × {the median absolute deviation VQSLOD of training set variants}. The transition/transversion SNP ratio suggested that the chosen cutoff criterion gave the best balance between selectivity and sensitivity.

SNPs call sets for six groups of *Ovis* and *Capra* animals were generated (i.e., Iranian and Moroccan domestics, and wilds for each genus). Because the analyses performed in this study required inter-group comparisons, we created genotype call sets at a consistent set of SNP sites for all animals from any group. For each genus, we merged the variant call sites from its three groups, and only retained biallelic positions without missing data. Genotypes were re-called at each biallelic SNP site for all individuals of interest by GATK UnifiedGenotyper using the option GENOTYPE_GIVEN_ALLELES. At this stage, the list of individuals was expanded to include the animals belonging to the world breed panels of sheep and goat (wpOA and wpCH) and additional wild samples that became available at this stage (4 *O. orientalis* and 4 *C. aegagrus*). Genotypes were improved and phased within groups by Beagle 4[53], and then filtered out where the genotype probability was less than 0.95. Finally, we filtered out sites that were monomorphic across the different subsets of individuals used in this study (see below).

In order to compare the signals of selection detected between *Ovis* and *Capra*, we performed a cross-alignment between the two reference genomes. First, we used the pairwise alignment pipeline from the Ensembl release 69 code base[54] to align the reference genomes of sheep (OARv3.1) and goat (CHIR1.0). This pipeline uses LastZ[55] to align at the DNA level, followed by post-processing in which aligned blocks are chained together according to their location in both genomes. The LastZ pairwise alignment pipeline is run routinely by Ensembl for all supported species, but the goat is not yet included in Ensembl. To avoid bias toward either species, we produced two different inter-specific alignments. One used sheep as the reference genome and goat as non-reference while the other used goat as the reference genome and sheep as non-reference. The difference is that genomic regions of the reference species are forced to map uniquely to single loci of the non-reference species, whereas non-reference genomic regions are allowed to map to multiple locations of the reference species. We obtained for segments of chromosomes of one reference genome the coordinates on the non-reference genome. Finally, for the SNPs discovered in one genus, we used the whole genome alignment with the reference genome of the other genus to identify the corresponding positions (Supplementary Table 6).

**Genetic structure.** In order to describe the genetic diversity within groups, we used VCFtools[56] to calculate genetic variation summary statistics on the 73 individuals for *Ovis* (i.e., 13 IROO, 20 IROA, 20 MOOA, and 20 wpOA) and 72 individuals for *Capra* (i.e., 18 IRCA, 20 IRCH, 20 MOCH, and 14 wpCH). The statistics measured were the total number of polymorphic variants (S) for the whole set of individuals in each genus and within each group, the averaged nucleotide diversity ($\pi$) within each group and the inbreeding coefficient (F) for each individual. Within each genus, the differences between the wild group and each domestic group were tested using a one-sided *t*-test for individual inbreeding and genetic load values, and a two-sided Mann–Whitney test for nucleotide diversity per site.

The overall divergence between the four groups within each genus (i.e., wild, Iranian and Moroccan domestics, and world panel) was estimated using all biallelic SNPs and the average weighted pairwise *Fst* following Weir and Cockerham[57] as implemented in VCFtools[56]. The genetic structure among groups was assessed with the clustering method sNMF[26], after pruning the data set to remove SNPs with linkage disequilibrium ($r^2$) greater than 0.2 using VCFtools. Linkage disequilibrium ($r^2$) was calculated between pairs of SNPs within sliding windows of 50 SNPs, with one SNP per pair randomly removed when $r^2$ was greater than 0.2. For each sNMF analysis, five runs of the same number of clusters (K) were performed with values of K from 1 to 10. We used the cross-entropy criterion to identify the most likely clustering solution, however, alternative partitions for different numbers of K were also explored to assess how individuals were divided between clusters.

To disentangle between shared ancestry and admixture, we ran TreeMix[27] to jointly estimate population splits and subsequent admixture events using the pruned data set used for sNMF. We ran TreeMix with the -global option to refine our maximum likelihood inferences. We rooted the TreeMix tree with the split between wild and domestic individuals. The block size for jackknifing was −k 500 SNPs, which approximately corresponds to 150 kb, exceeding the average blocks of LD found in both sheep and goats. We generated a Maximum Likelihood tree with no migration and then added migration events and examined the incremental change in the variance explained by the model and the residual values between individuals. The goal was to detect any potential high residual value or migration edge between wild and domestic individuals. To further explore the statistical relevance of possible admixture vectors identified by TreeMix (Supplementary Table 3), we calculated the three-population test f3[28] as a formal test of genetic introgression, using the qp3Pop program of the ADMIXTOOLS suite[58] for each combination of groups. For *Capra*, the wpCH group was divided between Australian breeds, French breeds, and Italian breeds. Results are reported in Supplementary Data 2.

**Demographic inference.** For each genus, we carried out ancestral demographic inference analyses using the MSMC model implemented in the MSMC2 software[25]. MSMC is based on the pairwise sequentially Markovian coalescent[59]; however, it uses haplotypes of phased genome sequence data as input. For each analysis we used two individuals from one group, thus 4 haplotypes. Each analysis was repeated for another random set of two individuals, i.e., a replicate of the analysis per group. Input and output files were generated and analyzed with the python scripts provided with the MSMC software and found at https://github.com/stschiff/msmc-tools. Analyses parameters were kept as default, except the mutation rate that was set to $2.5 \times 10^{-8}$ and the generation length was set to 2 years. In order to estimate the uncertainty on the time estimates, we varied these parameters (mutation rate of $2.5 \times 10^{-8}$ and $1.0 \times 10^{-8}$ in combination with generation length of 2 and 4 years)

and provided a rough estimate of the domestication period (see Supplementary Fig. 2).

**Genetic load**. Genetic load was estimated in two ways. Firstly, by calculating genetic load for each individual as the sum of deleterious fitness effects over all protein-coding genomic positions following the method of Librado et al.[60]. Briefly, as a proxy for evolutionary constraint, we used the PhyloP scores from the 46-way mammal alignment (http://hgdownload.cse.ucsc.edu/goldenPath/hg19/phyloP46way/placentalMammals/). From this alignment, we identified protein-coding sites evolving under functional constraints (phyloP score ≥1.5). For each *Ovis* or *Capra* genome, we then investigated whether these sites were mutated. If so, we summed the phyloP scores over all mutated sites, so that mutations in highly constrained sites contribute proportionally more to the total load estimate. This provided a load estimate for each sheep/goat genome. Finally, to obtain an average load per site, we divided by the total number of analyzed positions. It is worth noting that we conditioned on homozygous sites to avoid modeling the dominance coefficient of mutations at heterozygous sites (e.g., recessive, intermediate, dominant). Second, we compared gene-by-gene the genetic deleterious load in wild and domesticated *Ovis* groups by performing a Wilcoxon test, with the alternative hypothesis being that the domestic animals have more load than wild relatives. *p*-values were corrected for multiple testing[61] and we applied a threshold of adjusted *p*-values < 0.05. We performed a gene ontology enrichment analysis on the set of genes showing a significant increase in genetic load using WebGestalt[62,63]. As the reference genomes are poorly annotated for genes, we relied on single-copy orthologs between our species and human and mouse. Genes from the X chromosome were excluded from the background set. We did not carry out this analysis on *Capra* due to the higher inbreeding observed in the wild samples.

**Detection of selection signatures**. For detecting signatures of selection related to domestication, we used all the biallelic SNPs showing a minor allele frequency greater than 0.10 in at least one of the three groups tested (i.e., Iranian and the Moroccan domestic groups, and the wild group for each genus). Because we expected signatures of selection related to the domestication process to be present in all domestic animals, we adopted the following general strategy: we tested with hapFLK[29] (see Supplementary Note 5 and Supplementary Figures 9, 10 and 11) for each genus the wild group against each of the traditionally managed domestic groups (i.e., Iranian and Morocco) and focused on those common regions putatively under selection that were detected in both cases. Group sample sizes ($n =$ 13–20) were compatible with the requirements of the method[29]. We visually checked if the consistent signatures of selection found with hapFLK were also present in the corresponding world panel set of each genus, but did not include these groups in the statistical test due to their multi-breed composition. Finally, we looked for shared signals of selection between *Ovis* and *Capra* using a stratified FDR approach. The strategy is depicted in Supplementary Fig. 4.

We performed hapFLK tests for contrasting the wild group to each of the Iranian and Moroccan groups in each genus. The kinship matrix was calculated from the Reynold's genetic distances[64] between pairs of groups, using a random subset of one percent of the variants. The inferred population tree was built using the neighbor-joining algorithm. For each SNP, we performed the hapFLK test that incorporates haplotypic information to increase the power to detect selective sweeps. For each tested SNP, the hapFLK statistic calculated the deviation of haplotypic frequencies with respect to the neutral model estimated by the kinship matrix[65]. To exploit linkage disequilibrium information, hapFLK uses the Scheet and Stephens'[66] multipoint model for multilocus genotypes that can be fitted to unphased data. One of the main applications of this model is to perform phase estimation (fastPHASE software[66]). In our analysis, the model was trained on unphased data, and therefore our analysis accounts for phase uncertainty. The method was used to regroup local haplotypes along chromosomes in a specified number of clusters *K* set to 25, using a Hidden Markov Model.

To identify the common regions putatively under selection in the two traditionally-managed domestic groups for each genus, we combined the two previous hapFLK analyses. For each analysis the hapFLK scores were fitted to a $\chi^2$ distribution to obtain *p*-values (script available at https://forge-dga.jouy.inra.fr/projects/hapflk/documents). The results of the two contrasts between the wild group and each of the domestic groups were combined using Stouffer's method[67] to obtain single *p*-values for the comparison of wild vs. domestic animals. Finally, the FDR framework[68] was applied to the whole set of SNPs to convert the combined *p*-values into *q*-values. SNPs showing *q*-values < $10^{-2}$ were retained and grouped into genomic regions when they were less than 50 kb distant from each other.

To investigate whether the signal of selection was shared between *Ovis* and *Capra*, we first used the cross alignment of the two reference genomes to identify homologous segments. We then applied a stratified FDR framework[69]. This approach is based on the fact that there is an inherent stratification in the tests given the prior information in the genetic data[69], because the underlying distribution of true alternative hypotheses might be different according to the different dynamics of various genomic regions, leading to different distributions of *p*-values. This requires to obtain FDR adjusted *p*-values (i.e., *q*-values) separately for the different strata. We searched for convergences in each genus by separating the regions homologous to those detected in the other genus (referred as the shared

stratum) and the rest of the genome (referred as the general stratum). We extracted the *p*-values separately for each of the two defined strata and then calculated *q*-values through the FDR framework. These stratified *q*-values were the final quantities considered for statistical significance (<$10^{-2}$) to detect SNPs under selection and merge them into the corresponding genomic regions.

To test for convergent signatures of selection differentiating wild from domestic animals in both genera, we examined the relationship between the significance threshold applied to *q*-values (that we made vary from 0.2 to 0.002) in one genus and the estimated probability that a SNP is selected in the shared stratum of the other genus using Storey et al.[70] approach. An increase in the inferred probability with a decrease of the threshold applied to the *q*-value (increase in stringency) indicates that the more significant the region is in one genus, the more likely we would find significant SNPs in the other genus.

We filtered out the selection signals that were not consistent among the three domestic groups. For each detected region, we used the phased haplotypes of individuals which were clustered using Neighbor-Joining trees based on the percent of identity between sequences. Only regions showing consistent signals were kept (Supplementary Fig. 5).

In order to infer if the signals of selection detected with hapFLK indicated relaxation of selection or positive selection in the domestics, we estimated the difference in nucleotide diversity ($\pi$) on each putative region under selection between the wild and domestic groups. We expressed this difference as the $\Delta\pi$ index, which was calculated for each genomic region as the difference between $\pi$ calculated for the wild group and the average of $\pi$ for the Iranian and Moroccan domestic groups, minus the difference in $\pi$ between these two groups calculated over the whole genome:

$$\Delta\pi = (\pi_{\text{wilds}} - \pi_{\text{iran-morocco}})_{\text{genomic-region}} - (\pi_{\text{wilds}} - \pi_{\text{iran-morocco}})_{\text{whole-genome}}$$

A negative value would indicate that the nucleotide diversity is lower in the wild group compared to the average of the two domestic groups, and would be considered as showing a relaxation of selection in these last groups, diversifying selection in the domestics or positive selection in the wilds. Contrarily, a positive value would indicate directional positive or stabilizing selection that occurred in the domestic groups. We also used the haplotype clustering to manually verify in each region if the selective sweep detected confirmed the indications given by the $\Delta\pi$ index.

We conducted functional interpretations as follows. For each region under selection, we considered the region plus 50 kb on each side to identify functional roles and 5 kb upstream and downstream of genes and we assessed the overlap between these coordinates to retain the genes of interest. Finally we considered that a gene was related to a given detected region when the positions of the region and the gene were overlapping. We then assessed what gene was the most likely targeted by selection by considering the closest gene to the top signal, i.e., the position of the lowest *q*-value within the region. Genes were functionally annotated using Uniprot (http://www.uniprot.org/), by considering their involvement in 30 child terms (i.e., the terms' direct descents) of the "Biological Process" category (i.e., GO:0008150). We retrieved all GO terms corresponding to each gene (Supplementary Data 4) for 30 of the 33 categories, because we did not consider three terms that were not involved in mammalian functions (i.e., GO:0006791 sulfur utilization, GO:0006794 phosphorus utilization, GO:0015976 carbon utilization). We performed two $\chi^2$-tests to compare the distributions of genes in the GO categories, i.e., (i) genes under selection from genus-specific regions versus that from homologous regions, and (ii) all genes under selection versus the 18,689 human genes associated to GO terms in Swiss-Prot. In order to interpret genes functions in a livestock context, we also retrieved the information available from the literature on their phenotypic effects.

Finally, to find the SNPs within the previously detected regions that were the most differentiated between wild and domestic groups, we used the FLK statistic. As for hapFLK, it represents the deviation of single-marker allelic frequencies with respect to the neutral model estimated by the kinship matrix[65]. The same procedure was used to fit the scores from the two analyses to a $\chi^2$ distribution and combine the *p*-values obtained as was used for the hapFLK test. However, the non-uniform distribution of the *p*-values precluded applying the FDR framework and we selected SNPs within the regions detected with hapFLK showing *p*-values <$10^{-4}$. For these SNPs we used the Variant Effect Predictor (VEP) annotations[71] that were generated from the Ensembl v74 sheep OARv3.1 genome annotation for *Ovis* (http://www.ensembl.org/Ovis_aries/Tools/VEP) and from the goat CHIR1.0 genome annotation produced by the NCBI eukaryotic genome annotation pipeline for *Capra* (https://www.ncbi.nlm.nih.gov/genome/annotation_euk/process/). SNPs were classified as intergenic, upstream and downstream (including UTRs), and intronic and exonic positions. The differences between the distributions of SNPs with FLK *p*-values <$10^{-4}$ and all the SNPs used for detecting selection signatures were examined with a $\chi^2$-test.

**Data availability**. Sequences and metadata data generated for the 73 *Ovis* and 72 *Capra* samples used in these analyses are publicly available. General information and all vcf files can be found on the Ensembl website (http://projects.ensembl.org/nextgen/). All Fastq files, Bam files, and de novo assemblies of *O. orientalis* and *C. aegagrus* can be found on the European Nucleotide Archive (https://www.ebi.ac.uk/ena) under the accession code of the Nextgen project (PRJEB7436).

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

## Acknowledgements

This work was supported by the European Union 7th framework project NEXTGEN (Grant Agreement no. 244356, coordinated by P.T.), the FACCE ERA-NET Plus project CLIMGEN (grant ANR-14-JFAC-0002-01), the LabEx OSUG@2020 (Investissements d'avenir – ANR10LABX56), the Wellcome Trust [WT108749/Z/15/Z], and the European Molecular Biology Laboratory (I.S., L.Cl., P.F.). We thank Marina Naval-Sanchez for fruitful interactions. Maps were made using data freely distributed by the Land Processes Distributed Active Archive Center (LP DAAC), located at USGS/EROS, Sioux Falls, SD (http://lpdaac.usgs.gov).

## Author contributions

The paper represents the joint efforts of several research groups, most of whom were involved in the NEXTGEN project (coordinated by P.T.). P.T., F.P., E.C. and R.N. designed the study. P.T. and F.P. supervised the joint work in NEXTGEN. A.S., A.C., P.W., M.W.B., S.J., R.N., P.A.-M., H.R.R. and P.F. supervised the work of their research group. G.T.-K., H.R.R., S.N., B.B., W.Z. and A.C. collected the samples. A.S. and J.K. provided complementary whole-genome sequences. A.A. and S.E. conducted the laboratory work and produced whole-genome sequences. S.J. developed the sampling design and supervised Geographic Information Systems. I.S., L.Cl., E.C., S.E. and F.Bo. contributed to bioinformatic analyses. F.J.A, B.S., B.B., P.de.V., F.Bo., P.L., L.Co., F.Bi., F.P. and M.B. did the analyses. F.J.A, P.O.-t.W., F.Bo, F.P. and P.T. produced the figures. F.J.A, P.O.-t.W., M.W.B. and F.P. wrote the text with input from all authors and especially F. Bo., P.T. and L.O.

## Additional information

Florian J. Alberto[1], Frédéric Boyer[1], Pablo Orozco-terWengel[2], Ian Streeter[3], Bertrand Servin[4], Pierre de Villemereuil[1,21,21], Badr Benjelloun[1,5], Pablo Librado[6], Filippo Biscarini[7,22,22], Licia Colli[8,9], Mario Barbato[2,8], Wahid Zamani[1,23,23], Adriana Alberti[10], Stefan Engelen[10], Alessandra Stella[7], Stéphane Joost[11], Paolo Ajmone-Marsan[8,9], Riccardo Negrini[9,12], Ludovic Orlando[6,13], Hamid Reza Rezaei[14], Saeid Naderi[15], Laura Clarke[3], Paul Flicek[3], Patrick Wincker[10,16,17], Eric Coissac[1], James Kijas[18], Gwenola Tosser-Klopp[4], Abdelkader Chikhi[19], Michael W. Bruford[2,20], Pierre Taberlet[1] & François Pompanon[1]

[1]Univ. Grenoble Alpes, Univ. Savoie Mont Blanc, CNRS, LECA, F-38000 Grenoble, France. [2]School of Biosciences, Cardiff University, Museum Avenue, CF10 3AX Cardiff, Wales, UK. [3]European Molecular Biology Laboratory, European Bioinformatics Institute, Wellcome Genome Campus, Hinxton, Cambridge CB10 1SD, UK. [4]GenPhySE, INRA, INPT, ENVT, Université de Toulouse, 31326 Castanet-Tolosan, France. [5]Institut National de la Recherche Agronomique Maroc (INRA-Maroc), Centre Régional de Beni Mellal, Beni Mellal 23000, Morocco. [6]Centre for GeoGenetics, Natural History Museum of Denmark, Øster Voldade 5-7, 1350K Copenhagen, Denmark. [7]PTP Science Park, Bioinformatics Unit, Via Einstein-Loc. Cascina Codazza, 26900 Lodi, Italy. [8]Istituto di Zootecnica, Facoltà di Scienze Agrarie, Alimentari e Ambientali, Università Cattolica del S. Cuore, via Emilia Parmense n. 84, 29122 Piacenza (PC), Italy. [9]BioDNA - Centro di Ricerca sulla Biodiversità e DNA Antico, Facoltà di Scienze Agrarie, Alimentarie e Ambientali, Università Cattolica del S. Cuore, via Emilia Parmense n. 84, 29122 Piacenza (PC), Italy. [10]CEA - Institut de biologie François-Jacob, Genoscope, 2 rue Gaston Crémieux, 91057 Evry, France. [11]Laboratory of Geographic Information Systems (LASIG), School of Architecture, Civil and Environmental Engineering (ENAC), Ecole Polytechnique Fédérale de Lausanne (EPFL), 1015 Lausanne, Switzerland. [12]AIA Associazione Italiana Allevatori, 00161 Roma, Italy. [13]Laboratoire d'Anthropologie Moléculaire et d'Imagerie de Synthèse, CNRS UMR 5288, Université de Toulouse, Université Paul Sabatier, 31000 Toulouse, France. [14]Department of Environmental Sci, Gorgan University of Agricultural Sciences & Natural Resources, 41996-13776 Gorgan, Iran. [15]Environmental Sciences Department, Natural Resources Faculty, University of Guilan, 49138-15749 Guilan, Iran. [16]CNRS, UMR 8030, CP5706 Evry, France. [17]Université d'Evry, UMR 8030, CP5706 Evry, France. [18]CSIRO Agriculture, 306 Carmody Road, St. Lucia, QLD 4067, Australia. [19]Institut National de la Recherche Agronomique Maroc (INRA-Maroc), Centre Régional d'Errachidia, Errachidia 52000, Morocco. [20]Sustainable Places Research Institute, Cardiff University, 33 Park Place, CF10 3BA Cardiff, Wales, UK. [21]Present address: CEFE-CNRS, UMR 5175, 1919 route de Mende, 34293 Montpellier 05, France. [22]Present address: CNR-IBBA, Via Bassini 15, 20133 Milano, Italy. [23]Present address: Department of Environmental Sciences, Faculty of Natural Resources and Marine Sciences, Tarbiat Modares University, 46417-76489 Noor, Mazandaran, Iran. Florian J. Alberto, Frédéric Boyer, Pablo Orozco-terWengel and Ian Streeter contributed equally to this work.

