## [Peer Review File · Nature Communications]

REVIEWERS' COMMENTS:

Reviewer #1 (Remarks to the Author):

I am essentially satisfied with the current version of the manuscript and will recommend only minor modifications to remove remnants of the previous claims that were not strongly supported and that now have been justly removed from the current version. I would like to stress that the dataset provided here is highly relevant and that this manuscript is a worthwhile contribution to the genomics of domesticated animals.

The major modifications introduced now concern the assessment of the functional enrichment using a much less biased approach and the removal of the unconvincing claims. These modifications are appropriate and satisfactory. As discussed in my previous review, the literature-based search emphasizing a major role of genes linked to the nervous system was weak and revealed only preconceptual biases. For example, the global and ubiquitous chromatin component HMGI-C/HMGA2 cannot be univocally linked to the nervous system and the connection proposed was based on a 10-year-old publication showing that the knock-out of this gene in mice affected the Ink4/Arf expression in neural stem cells. This is, however, one of the many phenotypes of the deletion of a gene coding for a ubiquitous protein that does not even satisfy the author's claim that they enriched information contained in the gene ontology terms with specific phenotypes observed in livestock species. The association of this gene to the functional category of "neural system phenotypic class" should be deleted because it is not robust since it is common that mutations within genes with general functions affect also the nervous system. The association to categories based on the literature search is too biased and not strong enough as one can find any connection with the neural system for most of the sufficiently studied genes since about 80% of the genes can be found expressed in the brain. In general, the categories listed in table 1 are not convincing and the whole column should be deleted. References to the enrichment of genes involved in the nervous system that have been removed from most of the text should also be removed from the introductory paragraph.

Finally, the comparison between selection detection methods hapFLK and Fst/Pi that is useful to the readers could indeed be found as supplementary materials if the peer review process is published in its entirety on the website. I am concerned, however, that this would dilute this information with other points concerning weak claims that have now been removed from the final version. I believe that this interesting comparison would deserve a more prominent position within the supplementary result section.

Reviewer #2 (Remarks to the Author):

The authors have addressed all my concerns. Very interesting story and very useful data for the community.

Reviewer #1 (Remarks to the Author):

I am essentially satisfied with the current version of the manuscript and will recommend only minor modifications to remove remnants of the previous claims that were not strongly supported and that now have been justly removed from the current version. I would like to stress that the dataset provided here is highly relevant and that this manuscript is a worthwhile contribution to the genomics of domesticated animals.

The major modifications introduced now concern the assessment of the functional enrichment using a much less biased approach and the removal of the unconvincing claims. These modifications are appropriate and satisfactory. As discussed in my previous review, the literature-based search emphasizing a major role of genes linked to the nervous system was weak and revealed only preconceptual biases. For example, the global and ubiquitous chromatin component HMGI-C/HMGA2 cannot be univocally linked to the nervous system and the connection proposed was based on a 10-year-old publication showing that the knock-out of this gene in mice affected the *Ink4/Arf* expression in neural stem cells. This is, however, one of the many phenotypes of the deletion of a gene coding for a ubiquitous protein that does not even satisfy the author's claim that they enriched information contained in the gene ontology terms with specific phenotypes observed in livestock species. The association of this gene to the functional category of "neural system phenotypic class" should be deleted because it is not robust since it is common that mutations within genes with general functions affect also the nervous system. The association to categories based on the literature search is too biased and not strong enough as one can find any connection with the neural system for most of the sufficiently studied genes since about 80% of the genes can be found expressed in the brain.

Answer — We have removed the reference to "neural development" linked to HMGI-C in the main text, in the supplementary notes and in former Supplementary Table 7 (now Supplementary Data 3).

In general, the categories listed in table 1 are not convincing and the whole column should be deleted.

Answer — The column was deleted.

References to the enrichment of genes involved in the nervous system that have been removed from most of the text should also be removed from the introductory paragraph.

Answer — In the former "Introductory paragraph" (now "Abstract"), we would like to keep information about some functional categories associated to the genes under selection, independently of any enrichment. To avoid any confusion we replaced :

"Among the genomic regions carrying selective sweeps differentiating domestic breeds from wild populations, 20 were common to *Capra* and *Ovis*, and were associated to genes involved in nervous system, immunity and productivity traits."

by

"Among the genomic regions carrying selective sweeps differentiating domestic breeds from wild populations, which were associated among others to genes involved in nervous system, immunity and productivity traits, 20 were common to *Capra* and *Ovis*."

The genes related to nervous system here correspond to NBEA and MTMR7 genes that, according to Reviewer 1 "are expressed predominantly in the brain and would deserve the neural system annotation used here."

Finally, the comparison between selection detection methods hapFLK and Fst/Pi that is useful to the readers could indeed be found as supplementary materials if the peer review process is published in its entirety on the website. I am concerned, however, that this would dilute this information with other points concerning weak claims that have now been removed from the final version. I believe that this interesting comparison would deserve a more prominent position within the supplementary result section.

Answer — We added this comparison as Supplementary Note 5, which is illustrated by 3 new supplementary figures (Suppl. Fig 9 – 11).

Reviewer #2 (Remarks to the Author):

The authors have addressed all my concerns. Very interesting story and very useful data for the community.